# Childhood Emotional Abuse, Neuroticism, Perfectionism, and Workaholism in an Italian Sample of Young Workers

**DOI:** 10.3390/bs14040298

**Published:** 2024-04-04

**Authors:** Valeria Verrastro, Francesca Cuzzocrea, Danilo Calaresi, Valeria Saladino

**Affiliations:** 1Department of Health Sciences, University “Magna Graecia” of Catanzaro, 88100 Catanzaro, Italy; valeriaverrastro@unicz.it (V.V.); danilo.calaresi@unicz.it (D.C.); 2Department of Human Sciences, Society and Health, University of Cassino and Southern Lazio, 03043 Cassino, Italy; v.saladino@unicas.it

**Keywords:** childhood emotional abuse, neuroticism, perfectionism, workaholism, gender differences, emerging adults

## Abstract

The literature has linked childhood emotional abuse (CEA) to severe negative outcomes such as the development of several maladaptive personality traits and coping mechanisms. Nonetheless, its concurrent connection with neuroticism, perfectionism, and workaholism has not been explored. For the above reasons, the present study sought to investigate whether neuroticism and perfectionism mediate the relationship between CEA and workaholism, as well as evaluate the gender invariance of the model. The sample of the present research comprised 1176 young workers (50% women), aged 18–25, who completed validated self-report questionnaires. The findings highlighted significant positive direct and indirect paths, suggesting a complex interplay between CEA, neuroticism, perfectionism, and workaholism. Furthermore, the model exhibited no significant differences between genders, suggesting that the identified relationships are consistent across both women and men. The findings highlight the importance of identifying CEA and considering the adoption of trauma-informed approaches to manage its adverse effects, thereby potentially averting the onset of workaholism. Moreover, the results underline the necessity for customized preventive measures, aiming to mitigate traits associated with neuroticism and perfectionism as potential paths for successful therapeutic interventions.

## 1. Introduction

Childhood emotional abuse (CEA) represents a distressing and detrimental type of mistreatment that can exert enduring impacts on individuals’ overall well-being, and involves neglecting a child’s emotional requirements through behaviors and language that undermine their feelings of security, self-esteem, and emotional development [1]. Furthermore, CEA does not include physical violence, but can comprise verbal abuse and other behaviors that minimize the child’s personal value. A child can potentially experience CEA in multiple ways, such as when their emotional needs are ignored, their feelings are disregarded, they are kept from interacting with other family members and peers, or there is an atmosphere of fear and intimidation [1,2]. CEA can have enduring effects on an individual’s psychological balance [2], considering that it is directly linked to several negative outcomes such as neuroticism [3], perfectionism [4], and workaholism [5].

Neuroticism is a personality trait which implies a predisposition towards negative emotions and maladaptively shapes an individual’s responses to daily life events [6]. Recent research has discovered a direct link between CEA and neuroticism in adulthood [7], as it appears that individuals who have undergone emotional abuse during their developmental years are more vulnerable to displaying higher degrees of neurotic traits when compared to individuals who did not experience such abuse. Indeed, CEA can disrupt the formation of adaptive coping mechanisms, hinder the development of emotional regulation abilities, as well as distort an individual’s self-perception and worldview [1]. These experiences can induce a pervasive feeling of insecurity and fear, potentially giving rise to neurotic features [3]. In addition, the influence of CEA on neuroticism potentially extends beyond its direct effects. In fact, emotional abuse may also shape the development of attachment styles and interpersonal connections, thereby amplifying neuroticism [7]. Specifically, it seems that individuals who have experienced CEA may encounter challenges related to trust and intimacy, resulting in difficulties when it comes to forming and sustaining healthy relationships [1].

Research on the link between CEA and perfectionism has been growing in the last decades. Perfectionism can be viewed as a characteristic encompassing stringent standards, self-evaluation, and a quest for flawlessness [8]. While some degree of perfectionism may offer advantages, individuals who have experienced CEA might be prone to developing exceedingly high levels of perfectionism, which can significantly affect their overall welfare [9]. Indeed, the literature has highlighted a direct connection between CEA and the rise and maintenance of perfectionistic traits [4]. Arguably, it appears that individuals who have experienced CEA may adopt perfectionistic features as a means of coping, hence the strive for perfection may serve as a strategy to restore a sense of self-value and as a way to guard against potential emotional damage [10]. Moreover, high degrees of perfectionism may lead to negative outcomes such as increased stress, anxiety, and diminished overall well-being [11]. It appears that CEA may foster the development of two distinct dimensions of perfectionism: self-oriented perfectionism and socially prescribed perfectionism [9]. The former implies the establishment of excessively high personal standards, engagement in self-criticism and the experiencing of a deep feeling of failure or disappointment every time such standards are not met. On the other hand, the latter stems from the perception of external expectations of perfection, together with an excessive desire for external validation and approval.

Relevant research on workaholism has also been focused on its connection with individual antecedents such as CEA. Workaholism can be defined as an overwhelming compulsion to work at the cost of other aspects of life and personal wellbeing [12]. The literature indicates a direct connection between CEA and workaholism, emphasizing the importance of deepening the knowledge on the dynamics between early life experiences and work-related behaviors during adulthood [5]. Emotional abuse has the feature of influencing an individual’s self-perception, plausibly leading to a strong desire for external validation and approval. In this context, workaholism may potentially act as a sort of compensatory mechanism to counter the feeling of inadequacy, by seeking control, accomplishment, and validation through work-related achievements [13]. In addition, workaholism can function as a way to divert or avoid emotional anguish and sufferance linked to CEA, granting a momentary sense of purpose [5]. The link between CEA and workaholism, moreover, can be shaped by a range of other factors, encompassing individual traits, coping strategies, and environmental influences [13]. Individuals who have encountered emotional abuse during their childhood may in fact develop a strong attachment to work if they envision it as a central aspect of their identity and a primary source of their self-esteem [14]. Additionally, workaholism can plausibly be maintained by societal and cultural expectations which prioritize quantity over quality as indicators of success [12].

Research has aimed to develop a comprehensive model of workaholism by evaluating and synthesizing the literature from recent decades on this topic [15]. Specifically, the authors proposed several individual antecedents based on evidence suggesting that personal characteristics play a significant role in fueling workaholism [16], arguing that, with respect to personality traits, there exists both theoretical and empirical support for a direct link between workaholism and certain individual antecedents such as neuroticism (e.g., [17]), perfectionism (e.g., [18]), and emotional-related issues (e.g., [19,20]). Ultimately, such a comprehensive theoretical model posits that emotional-related issues, indicating a reduced ability to regulate negative emotions, may potentially lead to the emergence of maladaptive personality traits, which in turn may contribute to the development of workaholic behaviors [15]. It is thus reasonable to argue that neuroticism may play a significant role in workaholism by intensifying the negative emotional effects associated with CEA. Individuals with high levels of neuroticism may hence be more prone to experiencing heightened levels of anxiety, stress, and dissatisfaction, which can drive their relentless pursuit of work-related goals as a means of managing or avoiding these adverse emotions. Moreover, CEA can cultivate a mindset of perfectionism, where individuals establish excessively high standards for themselves and engage in ceaseless work efforts to meet or surpass those standards. Consequently, this maladaptive perfectionism may become intertwined with workaholism, as individuals may persistently strive for success, recognition, and validation in their professional endeavors.

Another important topic discussed in the relevant literature is the potential presence of gender differences in workaholism. Aziz and Cunningham [21] discovered that work stress and work–life imbalance are related to workaholism, regardless of gender. They also found that gender does not moderate the relationship between workaholism, work stress, and work–life imbalance. However, Burke and Berge Matthiesen [22] found that women tend to have higher scores in terms of feeling driven to work, negative effects, exhaustion, and professional efficacy, while scoring similarly to men in terms of experiencing flow at work and absenteeism. Additionally, Burgess et al. [23] found that men tend to score higher on work involvement and feeling driven to work, while women tend to score higher on job stress. Both genders showed similar scores on work outcomes.

CEA is a significant traumatic experience that can have long-lasting psychological effects; hence, investigating its link to workaholism provides valuable insights into how CEA impacts individuals’ work-related attitudes and behaviors. Neuroticism and perfectionism are personality traits associated with both CEA and workaholism, and they may serve as mediators in the relationship between the two. This suggests that CEA can influence work-related behaviors through these traits. Understanding these connections has practical implications for interventions targeting workaholism. It stresses the significance of trauma-informed methodologies that tackle the fundamental psychological injuries and offer suitable assistance. By acknowledging the impact of CEA and focusing on neuroticism and perfectionism as intermediary factors, interventions can effectively diminish the likelihood of workaholism and encourage more beneficial work practices. Moreover, by investigating the connections among the variables under study, it becomes feasible to refine current theories and frameworks in the domain of workaholism and enhance our comprehension of the intricate interaction between childhood experiences, personality traits, and outcomes related to work. Moreover, an extensive review of existing literature revealed a distinct lack of studies that simultaneously examine the relationship between CEA, neuroticism, perfectionism, and workaholism. Therefore, our study aims to fill this void by investigating the intricate dynamics among these variables. By explicitly highlighting this gap and articulating the pioneering role of our research in addressing it, we seek to contribute novel insights that advance understanding in the field and pave the way for future investigations. Lastly, despite previous research, there is still a limited understanding of the presence and intricacies of gender differences concerning workaholism and other work-related factors. To fill this gap in knowledge, further investigation is necessary to gain a more comprehensive understanding of the complex interplay between gender, work-related behaviors, attitudes, and outcomes.

Therefore, this study seeks to address these gaps in the existing literature. The primary objective of this study was to assess whether neuroticism and perfectionism parallelly mediate the relationship between CEA and workaholism (Figure 1). Hence, CEA was regarded as the predictor variable, while neuroticism and perfectionism were considered mediator variables, and workaholism was treated as the outcome variable. Specifically, we hypothesized that individuals reporting higher levels of CEA would demonstrate elevated levels of neuroticism and perfectionism, which, consequently, would be linked to increased levels of workaholism. We thus hypothesized the following:

(1)Neuroticism would mediate the relationship between CEA and workaholism.(2)Perfectionism would mediate the relationship between CEA and workaholism.

Additionally, as an exploratory analysis, this study also examined whether the proposed model was invariant across different genders.

## 2. Materials and Methods

### 2.1. Participants

This study involved a sample of 1176 young workers from Italy (50% women), with age ranging from 18 to 25 (mean age = 21.42, SD = 2.28). The participants were recruited through online platforms, primarily using social networks. In terms of educational background, 16% had completed middle school, 50% held a high school diploma, 31% had a university degree, and 3% had obtained a postgraduate degree. Concerning occupational status, 69% were employed, while 31% were self-employed. Lastly, regarding marital status, the distribution among the participants was as follows: 41% were single, 44% were engaged, 12% were cohabiting, and 3% were married.

### 2.2. Procedures

This study adhered to the ethical guidelines outlined in the Helsinki Declaration and the Italian Association of Psychology (AIP). Approval for the study was obtained from the Institutional Review Board of the Institute for the Study of Psychotherapy, School of Specialization in Brief Psychotherapies with a Strategic Approach (reference number: ISP-IRB-2023-4). Participants were invited to participate in an extensive online survey, each answer was set as mandatory, and there were no missing data. Only participants who provided informed consent were included in the study, and their participation was voluntary, without any form of compensation. The privacy and confidentiality of the participants were given utmost priority during all phases of the research.

### 2.3. Measures

#### 2.3.1. Childhood Emotional Abuse

The retrospective assessment of adolescents’ perceived emotional abuse during childhood was conducted using the CEA subscale of the Childhood Trauma Questionnaire—Short Form (CTQ-SF, [2]). The CTQ-SF CEA subscale has been demonstrated to possess strong validity in Italian individuals, as noted in previous research [24]. Participants were asked to indicate the extent of emotional abuse they recall experiencing during their childhood (e.g., “People in my family said hurtful or insulting things to me”). Each item was rated on a 5-point Likert scale, ranging from 1 (never true) to 5 (very often true). The scores for the five items were averaged, with higher scores indicating a higher degree of CEA. In the current study, the internal consistency was good (Table 1). The CTQ-SF is one of the most used instruments to assess childhood traumatic experiences, with satisfactory levels of convergent and discriminant validity, as well as high levels of sensitivity and specificity for all subscales [2,25].

#### 2.3.2. Neuroticism

Neuroticism was assessed with the Neuroticism subscale of the Italian version of the Big Five Inventory (BFI-N; [26,27]). The BFI-N consists of 8 items such as “I see myself as someone who can be tense”. Participants were asked to rate each item on a 5-point Likert scale, ranging from 1 (strongly disagree) to 5 (strongly agree). A higher score on the BFI-N indicates a higher level of neuroticism. In the current study, the internal consistency was good (Table 1). The BFI is a well-known instrument to assess personality traits with sound psychometric properties, including high internal consistency reliability, well-established factor structure, and strong convergent and discriminant validity [28,29].

#### 2.3.3. Perfectionism

Perfectionism was examined with the Italian version of Short Almost Perfect Scale (SAPS; [30,31]). Only the Discrepancy subscale was used, which comprises 4 items specifically designed to measure maladaptive perfectionism traits such as “I am hardly ever satisfied with my performance”. Participants were asked to rate each item on a 7-point Likert scale, ranging from 1 (strongly disagree) to 7 (strongly agree). Higher scores on the SAPS indicate higher levels of perfectionism. In the current study, the internal consistency was good (Table 1). The SAPS demonstrated good internal consistency, as well as satisfactory convergent, discriminant, and criterion-related validity in several studies (e.g., [30,31]).

#### 2.3.4. Workaholism

To measure workaholism, the Italian version of the Bergen Work Addiction Scale (BWAS; [13,32]) was employed. The BWAS is a self-report questionnaire comprising 7 items specifically designed to assess workaholic behaviors. Participants indicate their level of agreement with each item on a 5-point Likert scale, ranging from 1 (never) to 5 (always). Sample items include “How often during the last year have you thought of how you could free up more time to work?” Higher scores on the BWAS indicate higher levels of workaholism. In the current study, the internal consistency was good (Table 1). The BWAS has demonstrated robust psychometric characteristics, comprising elevated levels of internal consistency reliability, a reliable factor structure, and robust convergent and discriminant validity (e.g., [13,32]).

### 2.4. Statistical Analyses

IBM SPSS 27 and R Studio 2023.09.1 +494 with the lavaan package for R were used to conduct data analysis.

In order to check potential concerns about common method bias, an exploratory factor analysis (EFA) was conducted using Harman’s single-factor test, in which all variables were included. Harman’s single-factor test tests the amount of variance explained by the single factor. If the variance explained is above the critical standard of 50%, it underlines the presence of common method bias [33].

Confirmatory factory analyses (CFAs) were conducted to assess whether the self-report instruments demonstrated internal validity concerning their assumed internal structures by verifying their fit to the data and the relationships between variables and their indicators [34].

Discriminant validity testing was also conducted, which evaluates whether the study variables are distinct from each other by assessing the degree to which they correlate with each other, compared to their respective indicators. Specifically, we tested factor correlation estimates and their confidence intervals, as well as nested models which compare the baseline model to a series of constrained models built by limiting each factor correlation separately [35].

Descriptive statistics and correlation analyses were performed for all the main variables.

Structural equation modeling (SEM) with latent variables was employed to examine the mediation model. In this model, CEA served as the predictor variable, while neuroticism and perfectionism acted as mediators, and workaholism was the outcome variable. Latent variables are variables that cannot be directly observed or measured and are inferred from observed variables or indicators. We used a parceling approach to obtain the indicators of the latent variables present in our model (e.g., [36]). Furthermore, we aggregated the items from the questionnaire in three indicators of each latent variable, which is considered to be an optimal strategy compared to models based on observed variables in terms of model evaluation (e.g., [36]). Solutions were generated based on maximum-likelihood estimation. To assess the significance of the indirect effects, the bootstrap-generated bias-corrected confidence interval approach with 5000 resamples was utilized. Furthermore, we controlled for the effects of background variables (age and gender) by including them as predictors of all study variables.

Lastly, a multigroup path analysis (MGPA) was performed, considering gender as the group variable, to examine potential variations in structural paths between men and women. More specifically, a constrained model, where the paths of the hypothesized model were fixed to be equal across both groups, was compared to an unconstrained model, where all paths were permitted to differ between the two groups. A chi-square difference test between the baseline and constrained models was employed to evaluate the equivalence of the models across groups.

## 3. Results

### 3.1. Common Method Bias

An exploratory factor analysis was performed using Harman’s single-factor test, which involved including all variables. The primary factor accounting for variance amounted to 30.84%, falling below the critical threshold of 50% [33]. This suggests that the questionnaires utilized in this study did not exhibit significant common method bias [33].

### 3.2. Confirmatory Factor Analyses

Confirmatory Factor Analyses (CFAs) corroborated the factor structure of the measures through an assessment of the measurement models and an examination of fit indices (Table 1). In particular, we examined their adequacy in fitting the data and the associations between variables and their respective indicators [34].

### 3.3. Discriminant Validity Testing

Discriminant validity testing confirmed that the study variables were distinct from each other (Table 2). Specifically, we examined the estimates of factor correlations along with their confidence intervals, which were far below the cut-off criterion of 0.85 [35]. Additionally, we evaluated nested models that contrasted the baseline model with a set of restricted models created by individually constraining each factor correlation, whose chi-square significance supported discriminant validity [35].

### 3.4. Descriptive Statistics and Correlations

The descriptive statistics and correlations between the variables analyzed in the study are displayed in Table 3. The means identified in the present research align with the values reported in previous studies conducted by Kuo et al. [37], Gegieckaite and Kazlauskas [38], Tóth et al. [39], and Morkevičiūtė and Endriulaitienė [40]. Considering that the values obtained for kurtosis and skewness of the study variables were far below the suggested +2/−2 range, all variables appeared to be normally distributed [41]; thus, an adequate normal distribution was expected [42]. Furthermore, as far as regression and mediation analyses are concerned, the central limit theorem (CLT) assumes that, with a moderately high sample size, with reports as low as >200 [43], the distribution of residuals in the data will approximate ever greater normality [44,45].

### 3.5. Mediation Model

The proposed model was assessed using SEM with latent variables (Figure 2). The findings indicated an adequate fit (*χ*^2^(64) = 309.81, *p* < 0.001, comparative fit index (CFI) = 0.97, root-mean-square error of approximation (RMSEA) = 0.06 (90% confidence interval [CI] = 0.05–0.06), and standardized root mean square residual (SRMR) = 0.04). Although the chi-square was statistically significant, it is widely acknowledged that the chi-square test is extremely influenced by bigger sample sizes (e.g., [46]). Therefore, additional fit indices beyond the chi-square, such as the CFI, RMSEA, and SRMR, were considered (e.g., [47]).

The standardized factor loadings, reflecting the associations between each indicator and its corresponding latent variable, varied from 0.77 to 0.89. This range suggests that all latent constructs were effectively captured by their respective indicators.

Significant direct and indirect paths were observed among all the variables examined, with a statistical significance equal to or lower than 0.001 (Table 4). Specifically, as far as direct paths are concerned, CEA exhibited significant positive correlations with neuroticism (β = 0.41, *p* < 0.001), perfectionism (β = 0.48, *p* < 0.001), and workaholism (β = 0.18, *p* < 0.001). Moreover, we identified direct and positive relationships between neuroticism and workaholism (β = 0.21, *p* < 0.001), as well as between perfectionism and workaholism (β = 0.24, *p* < 0.001). Furthermore, delving into the indirect effects, we found noteworthy results. Neuroticism displayed a significant indirect effect in the pathway between CEA and workaholism (β = 0.09, *p* < 0.001), suggesting that neuroticism may serve as a mediator between childhood emotional abuse and workaholic tendencies. Similarly, perfectionism exhibited a significant indirect effect in the association between CEA and workaholism (β = 0.11, *p* < 0.001), indicating its potential role as a mediator in this relationship.

Regarding neuroticism and perfectionism, the R-squared values were 0.22 and 0.24, indicating that 22% and 24% of their variance, respectively, was accounted for by CEA. As for workaholism, the R-squared value was 0.31, suggesting that 31% of its variance was explained by the predictor and mediator variables (Figure 2).

### 3.6. Moderating Role of Gender

To explore potential variations in structural paths between men and women, a multigroup path analysis (MGPA) was conducted on the proposed model. Initially, a constrained model was tested, where the paths of the hypothesized model were set equal across both groups. The constrained model yielded *χ*^2^(96) = 283.46, *p* < 0.001, and CFI = 0.97. Subsequently, an unconstrained model was tested, allowing all paths to vary between the two groups. The unconstrained model produced *χ*^2^(96) = 292.31, *p* < 0.001, and CFI = 0.97. The fit indices of the unconstrained model did not significantly differ from the constrained model, indicating structural equivalence across both groups (Δ*χ*^2^(5) = 7.77, *p* = 0.17, ΔCFI < 0.001). Thus, the relationships were found to be comparable between men and women.

## 4. Discussion

The primary objective of this study was to examine the potential mediating role of neuroticism and perfectionism in the relationship between CEA and workaholism. All study variables were assessed using self-report instruments, and CEA was evaluated retrospectively. The findings of this study indicate that neuroticism and perfectionism do indeed mediate the association mentioned above. These results carry significant implications for enhancing our understanding of the underlying mechanisms that contribute to the development of workaholism in individuals.

The connection between CEA and maladaptive personality traits seems to be intricate. Previous research has underlined a direct correlation between CEA and neuroticism, suggesting that individuals experiencing CEA possess higher neurotic features when compared to those who have not experienced such abuse [3]. Such a link can be attributed to the chronic emotional distress and trauma often associated with CEA, which may disrupt the healthy development of emotional regulation and may contribute to greater emotional sensitivity and instability [48]. The emotional processing of individuals marked by CEA is influenced, leading to the possible formation of neurotic beliefs and coping strategies. Common emotional patterns in individuals experiencing CEA are in fact negative self-perceptions, low self-esteem, and a heightened vulnerability to stressful events [49]. The aforementioned patterns can foster the manifestation of neuroticism, as well as anxiety, worry, and a general emotional volatility [3]. Additionally, CEA can have long-lasting effects on neural pathways and brain structures involved in emotional and stressful responses. Neuroimaging studies have in fact identified alterations in the brain regions related to emotional processing and regulation among individuals who experienced CEA [50]. These neurobiological changes may, thus, further contribute to the development of neurotic traits and an overall higher emotional reactivity.

Evidence indicates a direct correlation between CEA and perfectionism. Indeed, it appears that individuals who have encountered CEA are more prone to displaying perfectionistic tendencies in comparison to those who have not experienced such abuse [4]. Perfectionism can function as a coping strategy for individuals who have undergone CEA. Perfectionistic tendencies may be developed by these individuals as a coping mechanism to restore control and self-worth [10]. In this way, striving for perfection can protect oneself from rejection and criticism while simultaneously fostering competence and self-worth. Moreover, those with high CEA levels can develop a need for outside validation to make up for not receiving enough emotional support and validation throughout their formative years [11]. In other words, perfectionism is generally driven by a desire to be recognized, approved, and accepted by others, as individuals believe that achieving perfection will provide greater affirmation. Furthermore, CEA can lead to the development of cognitive biases such as all-or-nothing thinking and excessive self-criticism [51]. These skewed habits of thinking can strengthen perfectionistic tendencies, since individuals believe that anything less than perfection is undesirable. They tend to view mistakes or imperfections as personal failures.

Consequently, maladaptive personality traits such as neuroticism and perfectionism may potentially contribute to the development of workaholism patterns in individuals who have experienced emotional abuse. Indeed, the findings of our study support this assertion, supported by compelling theoretical underpinnings and relevant research outlined below, suggesting that neuroticism and perfectionism may indeed act as mediators in this relationship.

As neurotic people have elevated levels of tension, worry, and self-doubt, they can frequently turn to work and success for comfort in an attempt to reduce these uncomfortable feelings. In this situation, research indicates that employment may act as a haven and a coping method, offering a brief solace from their unstable emotions [52]. Furthermore, those with high levels of neuroticism are more likely to suffer from burnout and stress at work, struggle to effectively handle demands at work, and experience emotional tiredness and dissatisfaction, as literature suggests [53]. Workaholism’s demanding nature, which is defined by its stress on long work hours, exacerbates these unfavorable effects and speeds up the development of burnout. Furthermore, studies show that neurotic people frequently turn to unhealthy coping mechanisms such as overcompensation or avoidance, which exacerbate the detrimental effects of workaholism [54]. Workaholism may thus seem as an overcompensation for persons with high neuroticism, motivated by an inbuilt need for accomplishment, affirmation, and control as a coping mechanism for emotional pain. Although immersing oneself in work could provide a brief reprieve from worry, this excessive concentration on work frequently results in the neglect of other important facets of life, as research indicates [55]. Therefore, for those who struggle with these difficult features, the entwinement of neuroticism and workaholism can generate a complicated web of psychological suffering, potentially sustaining a cycle of emotional turbulence and professional burnout.

The relevant literature, as underlined below, also supports the mediating role of perfectionism identified in our study’s analysis. Both perfectionism and workaholism are rooted in a shared drive for achievement and success. Perfectionists set exceptionally high standards and devote significant efforts to not only meet but surpass them in their professional endeavors. This relentless pursuit often manifests in an intense focus on work, characterized by prolonged hours and an unwavering commitment to flawless task completion, as evidenced in the literature [56]. Moreover, research suggest that perfectionists frequently grapple with a fear of failure or making mistakes, which fuels their inclination toward workaholism [57]. They may perceive excessive work as a means of sidestepping failure and believe that maintaining strict control over tasks will minimize the likelihood of errors or criticism, thereby reinforcing their predisposition toward workaholism. Additionally, perfectionists commonly subject themselves to significant self-imposed pressure to meet ideal standards, firmly believing that their self-worth hinges on achieving these benchmarks, consistent with existing studies [58]. Indeed, the literature suggests that this internalized pressure amplifies workaholic tendencies, as perfectionists struggle to strike a balance between work and their personal life [59]. In their relentless pursuit of perfection, they may prioritize work to such an extent that they inadvertently neglect personal well-being and social connections. This disregard for other aspects of life intensifies proclivities toward workaholism, as the relentless pursuit of perfection overtakes considerations for other domains, as indicated by the literature [55].

The potential mediating roles of neuroticism and perfectionism are also supported by the theoretical framework of Loscalzo and Giannini [15], which summarizes the last decades of studies and theories on workaholism. Indeed, the comprehensive theoretical model suggests that there may be some individual antecedents, such as personality traits and emotional-related issues, that potentially pave the way for the development and maintenance of workaholic behaviors. Specifically, research suggests that emotional wounds might lead to the development and solidification of negative personality traits, which could then exacerbate the emergence of workaholic behaviors as individuals may seek to alleviate their unsettling emotions through excessive work [15,17,18,19,20].

The second aim of this study was to examine whether the proposed model remained consistent across genders. The findings indicated that the structural relationships within the model were consistent and did not differ between men and women. This suggests that the mediating effects of neuroticism and perfectionism in the association between CEA and workaholism may hold equal significance for both genders. These results contribute to the generalizability of our findings across genders. Previous research has identified variations in the relationship between workaholism and other psychological factors when comparing boys and girls [21,22,23]. The results of this study indicate that variations in workaholism among genders can be partially explained by individual differences in CEA, neuroticism, and perfectionism.

## 5. Limitations and Future Directions

Our research has several limitations that need to be acknowledged. First, it is challenging to determine the causal direction of the associations identified due to the study’s cross-sectional methodology. Longitudinal studies might be beneficial in validating and elucidating these findings over an extended period. Furthermore, the utilization of self-reported data raises the potential for interpretation bias. The replies of participants may be influenced by their subjective interpretations and impressions. Future research may take into account combining data from other sources, such as objective measurements or accounts from other people, to lessen this bias and offer a more thorough evaluation of the factors being examined. Third, it is critical to remember that we only used online data collection for this investigation. Due to this, our findings might not have been as applicable to people without internet access or to those who are less inclined to participate in online surveys. It would be advantageous to use a variety of data sources, such as offline samples or in-person interviews, to increase the accuracy and representativeness of future studies. This would enable a more thorough comprehension of the phenomenon in many situations and people. Finally, the convenience sample of this study may reduce the generalizability of the findings. It would be advisable to use different sampling techniques in order to ensure potentially less biased results.

From a clinical perspective, the findings of this study suggest the potential significance of early detection and intervention for individuals who have experienced CEA. Mental health professionals may consider utilizing the proposed model cautiously as a tool to identify individuals who might be at risk of developing workaholism and consider implementing appropriate therapeutic strategies with care. It may thus be important to acknowledge the complexity of addressing the underlying emotional wounds associated with CEA to potentially mitigate the adoption of maladaptive coping mechanisms, including workaholism. Furthermore, it is important to approach these tactics cautiously and acknowledge the limits of the current research, even if the identification of neuroticism and perfectionism as possible mediators may provide routes for therapeutic intervention. Furthermore, the recommendation to use trauma-informed strategies in order to establish a secure and encouraging atmosphere for recovery and healing has to be carefully evaluated in light of the particular requirements and circumstances of every case.

Regarding research, this study’s mediation model emphasizes how crucial it is to carry out long-term studies as a means to determine the temporal relationships between the variables. By following individuals over an extended period, researchers can examine the long-term effects of CEA on the development of neuroticism, perfectionism, and, ultimately, workaholism. Furthermore, there is a need for intervention studies to evaluate the effectiveness of therapeutic approaches that specifically target neuroticism and perfectionism. Such studies would provide valuable insights into whether reducing these personality traits can help mitigate the risk of workaholism and guide the development of prevention strategies.

## 6. Conclusions

This study provides preliminary insights into the intricate dynamics between CEA, neuroticism, perfectionism, and workaholism. Nonetheless, while there is evidence hinting at potential mediation, as well as supporting theoretical frameworks and research, additional investigation is surely required to bolster the theoretical and statistical foundation of this study’s results. A deeper understanding is necessary to inform future research endeavors with the objective of mitigating the adverse consequences of CEA and its associated psychological factors on workaholism. Deeply comprehending these connections can serve as a valuable resource for clinical practitioners, providing guidance for effective interventions.

## Figures and Tables

**Figure 1 behavsci-14-00298-f001:**
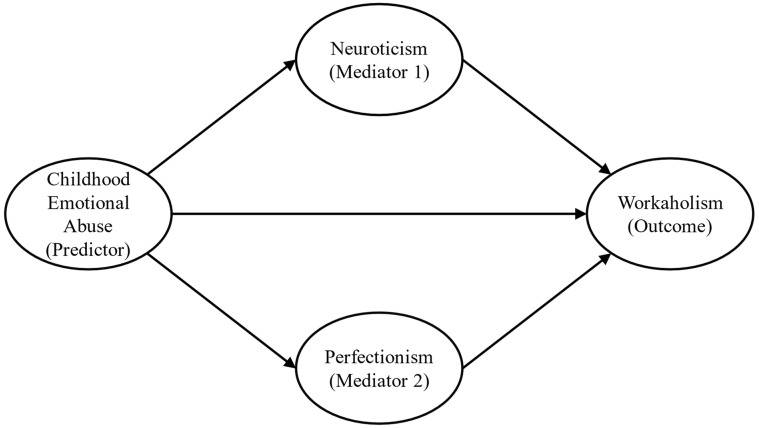
Hypothesized model.

**Figure 2 behavsci-14-00298-f002:**
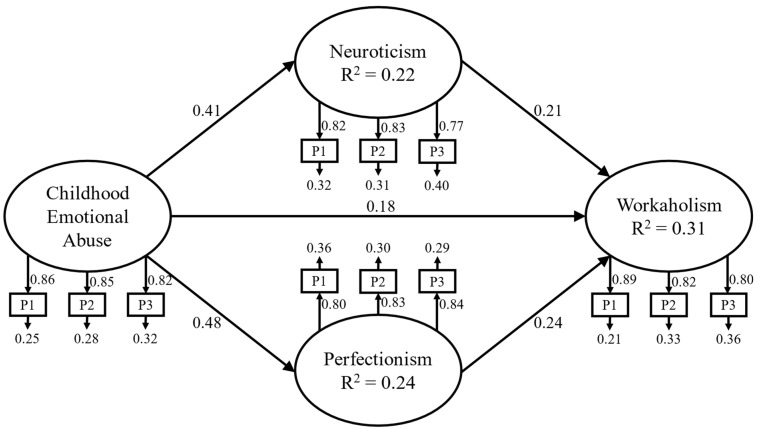
Structural Model. Note: P = parcel; only direct paths are reported for clarity purposes.

**Table 1 behavsci-14-00298-t001:** Goodness-of-fit indices of the measurement models.

	GFI	AGFI	NFI	CFI	RMSEA	SRMR	TLI
1. CEA	0.99	0.98	0.99	0.99	0.04	0.01	0.99
2. Neuroticism	0.98	0.95	0.97	0.98	0.07	0.03	0.96
3. Perfectionism	0.99	0.95	0.99	0.99	0.09	0.01	0.97
4. Workaholism	0.98	0.95	0.98	0.98	0.07	0.03	0.96

Note: CFI: comparative fit index; RMSEA: root-mean-square error of approximation; SRMR: standardized root-mean-square residual; GFI: goodness-of-fit index; AGFI: adjusted goodness-of-fit index; NFI: normed fit index; TLI: Tucker–Lewis Index.

**Table 2 behavsci-14-00298-t002:** Estimated factor loadings and chi-square difference tests for discriminant validity.

	FCE	*CI*	*CI*	*χ^2^*	*df*	Δ*χ^2^*	Δ*df*
		*LL*	*UL*				
1. CEA—Neuroticism	0.42	0.37	0.48	4028.30	249	1852.17 *	3
2. CEA—Perfectionism	0.47	0.42	0.53	3905.44	249	1729.32 *	3
3. CEA—Workaholism	0.34	0.28	0.39	4357.71	249	2181.59 *	3
4. Neuroticism—Perfectionism	0.59	0.54	0.64	3306.23	249	1130.11 *	3
5. Neuroticism—Workaholism	0.37	0.31	0.43	4141.45	249	1965.32 *	3
6. Perfectionism—Workaholism	0.41	0.36	0.47	3911.00	249	1734.88 *	3

Note: FCE: factor correlation estimates; *CI* = confidence interval; *LL* = lower limit; *UL* = upper limit; *χ*^2^ = chi-square; *df* = degrees of freedom; Δ = Delta; * *p* < 0.001.

**Table 3 behavsci-14-00298-t003:** Descriptive analyses and correlations.

	M	SD	Ske	Kur	α	1	2	3
1. CEA	1.95	0.99	0.99	0.00	0.87	-	-	-
2. Neuroticism	3.30	0.77	−0.33	−0.18	0.85	0.36 **	-	-
3. Perfectionism	4.35	1.52	−0.07	−0.91	0.86	0.40 **	0.51 **	-
4. Workaholism	2.64	0.98	0.02	−0.75	0.85	0.31 **	0.31 **	0.40 **

Note: *n* = 1176. ** *p* < 0.01. Ske = skewness. Kur = kurtosis.

**Table 4 behavsci-14-00298-t004:** Path Estimates, SEs and 95% CIs.

	*β*	*p*	*SE*	*CI*	*CI*
				*LL*	*UL*
Direct Effect					
Childhood Emotional Abuse → Neuroticism	0.41	<0.001	0.03	0.27	0.37
Childhood Emotional Abuse → Perfectionism	0.48	<0.001	0.05	0.60	0.78
Childhood Emotional Abuse → Workaholism	0.18	<0.001	0.04	0.11	0.27
Neuroticism → Workaholism	0.21	<0.001	0.06	0.18	0.41
Perfectionism → Workaholism	0.24	<0.001	0.04	0.11	0.25

Indirect Effect via Neuroticism					
Childhood Emotional Abuse → Workaholism	0.09	<0.001	0.06	0.06	0.13

Indirect Effect via Perfectionism					
Childhood Emotional Abuse → Workaholism	0.11	<0.001	0.03	0.07	0.17

Note: *p* = level of significance; *SE* = standard error; *CI* = confidence interval; *LL* = lower limit; *UL* = upper limit.

## Data Availability

The data presented in this study are available on request from the corresponding author.

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
