# Peer review of "Childhood Emotional Abuse, Neuroticism, Perfectionism, and Workaholism in an Italian Sample of Young Workers"

_behavsci, 2024, doi:10.3390/bs14040298_

Round 1

Reviewer 1 Report

Comments and Suggestions for Authors

The topic is relevant. The results of the research are important both scientifically and practically. The manuscript is quite good quality. I just have few comments. Please read comments below:

1) I propose to describe/explain the CEA concept more widely and deeper so that a reader who is not familiar with this construct can understand it.

2) You use retrospective method evaluating CEA. This should be indicated in the method section, also in Discussion.

3) In the description of CEA instrument you write  "The assessment of adolescents' CEA"  (line 170) and "the extent of CEA severity experienced during their childhood" (line 175). The description of the instrument needs clarification.

4) The text needs edition. Two sentences about the same in lines 178-179. You write "a 7-point Likert scale, ranging from 1 (strongly disagree) to 5 (strongly agree)" in line 194. Maybe it is better to write between men and women instead boys and girls (line 215)?

Reviewer 2 Report

Comments and Suggestions for Authors

This article explores the relationships between childhood emotional abuse (CEA), certain personality traits (neuroticism and perfectionism), and workaholism in young adults aged 18-25. Its main contribution appears to be the exploration of the associations among these constructs; however, the literature review is limited which makes it difficult to assess the originality of this study. Strengths of the study include a large sample of young adults with a balance gender representation and the test of invariance by gender. The statistical analysis (mediation model) may be a strength but is insufficiently explained.

Overall, the manuscript is relevant for the field and presented in a well-structured manner but have important weaknesses that are explained below.

General comments

Introduction

The introduction reviews studies on the relationships between CEA and neuroticism, CEA and perfectionism, as well as CEA and workaholism. However, it does not discuss studies that explored associations between these constructs altogether; no study in which CEA is linked to workaholism through neuroticism and perfectionism is presented. Are there previous studies that did this? Are there previous mediation studies? If so, what did they find? The introduction should include this information to support the scientific relevance of this study.

Lines 102-111 explain that neuroticism and perfectionism may play a role in workaholism among person with CEA. This is plausible, but the manuscript lacks a compelling theoretical framework and empirical evidence to justify the exploration of these relationships. A more robust theoretical and empirical foundation is necessary to support the exploration of this hypothesis in this paper.

Methods

Participants, procedures and measures are clearly outlined. A limitation, that is identified by authors, is the exclusive reliance on self-report measures.

The authors should provide details on the psychometric properties of the tools used, including validity and reliability, beyond just internal consistency in the sample.

As for the statistical analyses, the mediation model is insufficiently described. Authors refer to latent variables (l. 209) but do not explain what these variables are, how they were estimated and why they were preferred to observed variables. MGPA as well could be more thoroughly described. The statistical analyses section needs further elaboration for clarity and reproducibility.

Results

Overall, the mediation model results are insufficiently detailed, yet the gender moderation analysis is clearly presented. This section needs to be significantly expanded.

Descriptive statistics shows that the mean CEA score is 1.96 with a SD of 1.01. Considering that the CEA measure is five items rated on a 5-point Likert scale, ranging from 1 (never true) to 5 (very often true), the minimum score would be 5 and the maximum would be 25. How a mean of 1.96 was obtained? Is this the mean of the latent variable? The same logic applies to the other measures, where the means do not seem to correspond to the scales range. These apparent discrepancies make it difficult to assess if CEA exposition is present enough in this sample to achieve the aim of the study. If the exposition to CEA in the sample is low or very low, it limits any conclusions about Childhood Emotional Abuse.

Discussion

This section does not discuss the results of the mediation analysis. Major revisions are required to interpret these results within the context of existing research, exploring congruences or disparities and their implications. The invariance of the model across gender is however discussed adequately.

Conclusion

The conclusions lack clear support from the results, largely due to the underexplained findings. It is certainly premature to recommend TIC or interventions based on these results (e.g. l.361-364).

Specific comments

Many cited references are recent publications. The paper includes an acceptable number of self-citations.

The sentence from l.64-66 (Although…) should be supported by a reference.

l.155-156 ‘In relation to occupational status, 68% were employed, while 31% were self-employed.’ 1% not employed or is this an error? Please precise given that the participants are identified as ‘workers’

I have ethical concerns because ‘participants were invited to participate in an extensive online survey, and their completion of the survey was mandatory, ensuring no missing data. (l. 163-164).

Figure 1 is important and useful but is unclear. It is difficult to interpret and understand, probably because the mediation model is insufficiently described, as pointed above.

Reviewer 3 Report

Comments and Suggestions for Authors

The topic is exciting and relevant. However, the statistical analysis needs to be more robust.

Every time you discuss correlation (lines 45, 67, 87), you need to state whether it is direct or inverse.

Why don’t you present the hypotheses related to the correlation between variables presented in the literature? This is a better way to show the meaning of the arrows in your model.

Lines 152-3: “The participants were recruited through online platforms, primarily using social networks.” This is a convenience sample. The limitations of the study should mention this fact.

Do the variables follow a normal distribution?

Did you perform a confirmatory factor analysis?

What type of estimator was used with SEM?

Is there discriminant validity between the four measures?

How must total variance be explained?

Did you assess the problems related to the common variance method?

The interpretation of the chi-square needs to be corrected. Contrary to what you say, it indicates the model is very poor in fitness. Please study how to overcome this common problem.

The factor loadings presented are very low. They are significant, probably because of the size of the sample. What is the size of the effects?

Your results show that both direct and indirect effects exist. However, more is needed to conclude that there is a mediated effect. A partial possible mediate effect needs to be studied in other conditions to become more solid theoretically.

The data used in this study must be better analyzed, considering a deeper statistical survey of the distribution of variables and the adjusted method to estimate the factor loadings. I think one should assess their validity in the new sample whenever one uses measurement scales.

Round 2

Reviewer 3 Report

Comments and Suggestions for Authors

Congrats on your extensive revision of the paper, which has improved a lot.

Best wishes!